# Characterisation and comparison of semen microbiota and bacterial load in men with infertility, recurrent miscarriage, or proven fertility

**Shahriar Mowla[1†], Linda Farahani[2,3†], Tharu Tharakan[3,4], Rhianna Davies[3], Goncalo DS Correia[1,5], Yun S Lee[1,5], Samit Kundu[1,5], Shirin Khanjani[6], Emad Sindi[3], Raj Rai[2], Lesley Regan[2,7], Dalia Khalifa[8], Ralf Henkel[3,9,10], Suks Minhas[4], Waljit S Dhillo[3], Jara Ben Nagi[11], Phillip Bennett[1,5,7], David A MacIntyre[1,5,7*†], Channa N Jayasena[3,8*†]**

[1]Institute of Reproductive and Developmental Biology, Imperial College London, London, United Kingdom; [2]Wolfson Fertility Unit, Department of Gynaecology, St. Mary's Hospital, Imperial College NHS Trust, London, United Kingdom; [3]Section of Endocrinology & Investigative Medicine, Imperial College London, London, United Kingdom; [4]Department of Urology, Charing Cross Hospital, Imperial College NHS Trus, London, United Kingdom; [5]March of Dimes European Prematurity Research Centre, Imperial College London, London, United Kingdom; [6]Department of Gynaecology, University College London Hospital, London, United Kingdom; [7]Tommy's National Centre for Miscarriage Research, Imperial College London, London, United Kingdom; [8]Department of Andrology, Hammersmith Hospital, Imperial College NHS Trust, London, United Kingdom; [9]LogixX Pharma, Theale, Berkshire, United Kingdom; [10]Department of Medical Bioscience, University of the Western Cape, Bellville, South Africa; [11]Centre for Reproductive and Genetic Health (CRGH), London, United Kingdom

*For correspondence:
d.macintyre@imperial.ac.uk
(DAMacI);
c.jayasena@imperial.ac.uk (CNJ)

[†]These authors contributed equally to this work

## eLife Assessment

This **valuable** study reports a potential connection between the seminal microbiome and sperm quality/male fertility. The data are generally **convincing**. This study will be of interest to clinicians and biomedical researchers who work on microbiome and male fertility.

**Abstract** Several studies have associated seminal microbiota abnormalities with male infertility but have yielded differing results owing to their limited sizes or depths of analyses. The semen microbiota during recurrent pregnancy loss (RPL) has not been investigated. Comprehensively assessing the seminal microbiota in men with reproductive disorders could elucidate its potential role in clinical management. We used semen analysis, terminal-deoxynucleotidyl-transferase-mediated-deoxyuridine-triphosphate-nick-end-labelling, Comet DNA fragmentation, luminol reactive oxidative species (ROS) chemiluminescence, and metataxonomic profiling of semen microbiota by 16S rRNA amplicon sequencing in this prospective, cross-sectional study to investigate composition and bacterial load of seminal bacterial genera and species, semen parameters, ROS, and sperm DNA fragmentation in men with reproductive disorders and proven fathers. 223 men were enrolled, including healthy men with proven paternity (n=63), the male partners in a couple

encountering RPL (n=46), men with male factor infertility (n=58), and the male partners of couples with unexplained infertility (n=56). Rates of high sperm DNA fragmentation, elevated ROS, and oligospermia were more prevalent in the study group compared with control. In all groups, semen microbiota clustered into three major *genera*-dominant groups (1, *Streptococcus*; 2, *Prevotella*; 3, *Lactobacillus* and *Gardnerella*); no species clusters were identified. Group 2 had the highest microbial richness (p<0.001), alpha-diversity (p<0.001), and bacterial load (p<0.0001). Overall bacterial composition or load has not been found to associate with semen analysis, ROS, or DNA fragmentation. Whilst global perturbation of the seminal microbiota is not associated with male reproductive disorders, men with unidentified seminal *Flavobacterium* are more likely to have abnormal seminal analysis. Future studies may elucidate if *Flavobacterium* reduction has therapeutic potential.

## Introduction

Mean sperm counts reported within clinical studies have reduced annually by 2.6% since 2000 (*Levine et al., 2017*). Male factor accounts for approximately half of all cases of infertility, yet there are limited available interventions to improve sperm quality. Understanding the pathogenesis of male infertility may reveal novel therapeutic approaches for treating affected couples.

Symptomatic genitourinary infection is an established cause of male infertility, which may be detected by semen culture and treated with antibiotics (*Eini et al., 2021*; *Jung et al., 2016*). The current European Association of Urology guidance states that whilst antibiotics may improve overall semen quality, there is no evidence of increased pregnancy rates after antibiotic treatment of the male partner (*Brunner et al., 2019*; *Jung et al., 2016*). Seminal leucocytes release bactericidal reactive oxygen species (ROS) in response to infection; however, paradoxically, this may damage sperm DNA and impair semen quality (*Aitken et al., 2022*). We and others have reported that asymptomatic men affected by recurrent pregnancy loss (RPL), infertility, or impaired preimplantation embryo development have increased risks of high seminal ROS and sperm DNA fragmentation (*Benchaib et al., 2007*; *Jayasena et al., 2019*; *Aitken and Bakos, 2021*; *De Iuliis et al., 2009*; *Vorilhon et al., 2018*; *Agarwal et al., 2006*). It is therefore plausible that asymptomatic seminal infections may be associated with impaired reproductive function in some men. Since semen culture has a limited scope for studying the seminal microbiota due to its inability to identify all present microbiota, next-generation sequencing (NGS) approaches have been reported recently by a growing number of investigators (*Hou et al., 2013*; *Weng et al., 2014*; *Monteiro et al., 2018*; *Baud et al., 2019*; *Lundy et al., 2021*; *Garcia-Segura et al., 2022*; *Yang et al., 2020*). These studies, with varying methodologies, have produced inconsistent and conflicting results. As such, the composition of semen microbiota and its associations with clinical and molecular markers of male reproduction remain understudied. Elucidation of an association would have wide clinical application with therapeutic potential coupled with reproductive disorders (*Altmäe et al., 2019*).

We hypothesised that semen microbiota composition associates with functional semen parameters, including ROS levels and sperm DNA fragmentation. To test this, we explored relationships between metataxonomic profiles of bacteria, bacterial copy number, and key parameters of sperm function and quality in semen samples collected from 223 men, including those diagnosed with male factor infertility (MFI), unexplained infertility (UI), partners affected by recurrent miscarriage, and paternity-proven controls.

## Results
### Study population

Semen samples were collected from a total of 223 men; this included control (n=63) and a study group (n=160) comprised of men diagnosed with MFI (n=58), male partners of women with RPL (n=46), and male partners of couples diagnosed with UI (n=26). The overall mean age of the total cohort was 38.1±6 (mean ± SD). The mean age for controls was 40.1±8, and the mean age for patients undergoing various fertility investigations was 37±4.8. Ethnicity representation amongst recruited cohorts were not significantly different (p=0.38, chi-squared; *Appendix 1—table 1*).

**Table 1.** Patient demographics and notable parameters of seminal quality and function for controls and study subjects. Fisher's exact tests for all except age. Chi-squared test for age (n=223).

| Factor | Categories | Controls | Study cases | p-Value |
|---|---|---|---|---|
| DNA fragmentation index | Low | 45/114 (40%) | 69/114 (60%) | |
| | High | 12/82 (15%) | 70/82 (85%) | 0.0002*** |
| ROS | <3.77 RLU/s | 53/143 (37%) | 90/143 (63%) | |
| | >3.77 RLU/s | 5/33 (15%) | 28/33 (85%) | 0.02* |
| Semen volume | Optimal | 55/208 (26%) | 153/208 (84%) | |
| | Suboptimal | 8/15 (53%) | 7/15 (47%) | 0.03* |
| Age | <34 | 11/49 (22%) | 38/49 (88%) | |
| | 34–41 | 31/124 (25%) | 93/124 (85%) | |
| | >41 | 21/50 (42%) | 29/50 (58%) | 0.04* |
| Ethnicity | Caucasian | 39/156 (25%) | 117/156 (75%) | |
| | Non-Caucasian | 24/67 (36%) | 43/67 (64%) | 0.10 |
| Concentration | >15 M/ml | 58/182 (32%) | 124/182 (68%) | |
| | <15 M/ml | 5/41 (21%) | 36/41 (79%) | 0.01* |
| Progressive motility | >32% | 60/207 (29%) | 147/207 (71%) | |
| | <32% | 3/16 (19%) | 13/16 (81%) | 0.56 |
| Sperm morphology | >4% | 22/74 (30%) | 52/74 (70%) | |
| | <4% | 41/144 (28%) | 103/144 (72%) | 0.87 |
| Semen quality | Optimal | 24/78 (31%) | 54/78 (69%) | |
| | Suboptimal | 39/145 (27%) | 106/145 (73%) | 0.53 |

## Semen quality assessment

Rates of high sperm DNA fragmentation, elevated ROS, and oligospermia were more prevalent in the study group compared with control (*Table 1*). The study group represented 85% of samples with high sperm DNA fragmentation, 85% of samples with elevated ROS, and 79% of samples with oligospermia. Rates of abnormal seminal parameters including low sperm concentration, reduced progressive motility, and ROS concentrations were found to be highest in the MFI group (*Appendix 2—figure 1*). Baseline characteristics between the RPL, unexplained subfertility, and controls groups were similar (*Appendix 2—figure 1*).

## Seminal microbiota

Following decontamination, a total of 7,998,565 high-quality sequencing reads were identified and analysed. Hierarchical clustering (Ward's linkage) of relative abundance data resolved to genera level identified three major clusters, as determined by average silhouette score, amongst all samples (*Figure 1*, *Figure 1—figure supplement 1*). These were compositionally characterised by highest mean relative abundances of (1) *Streptococcus* (23.8%), (2) *Prevotella* (24.4%), or (3) *Lactobacillus* and *Gardnerella* (35.9% and 35.6%, respectively, *Figure 1C*). Assessment of bacterial load using qPCR showed Clusters 2 and 3 had significantly higher bacterial loads compared to Cluster 1. Similar analyses were performed using sequencing data mapped to species level; however, examination of individual sample silhouette scores within resulting clusters highlighted poor fitting, indicating a lack of robust species-specific clusters (*Figure 2—figure supplement 1*). To further investigate potential pairwise ecological interactions between taxa, a co-occurrence analysis was performed on the sequencing data, mapped to species level, with the SparCC algorithm (*Figure 2*). Five major graph communities were detected. Community 1 highlighted a co-occurrence pattern between *Gardnerella vaginalis* and *Lactobacillus iners,* in agreement with the composition of Cluster 3 from the hierarchical

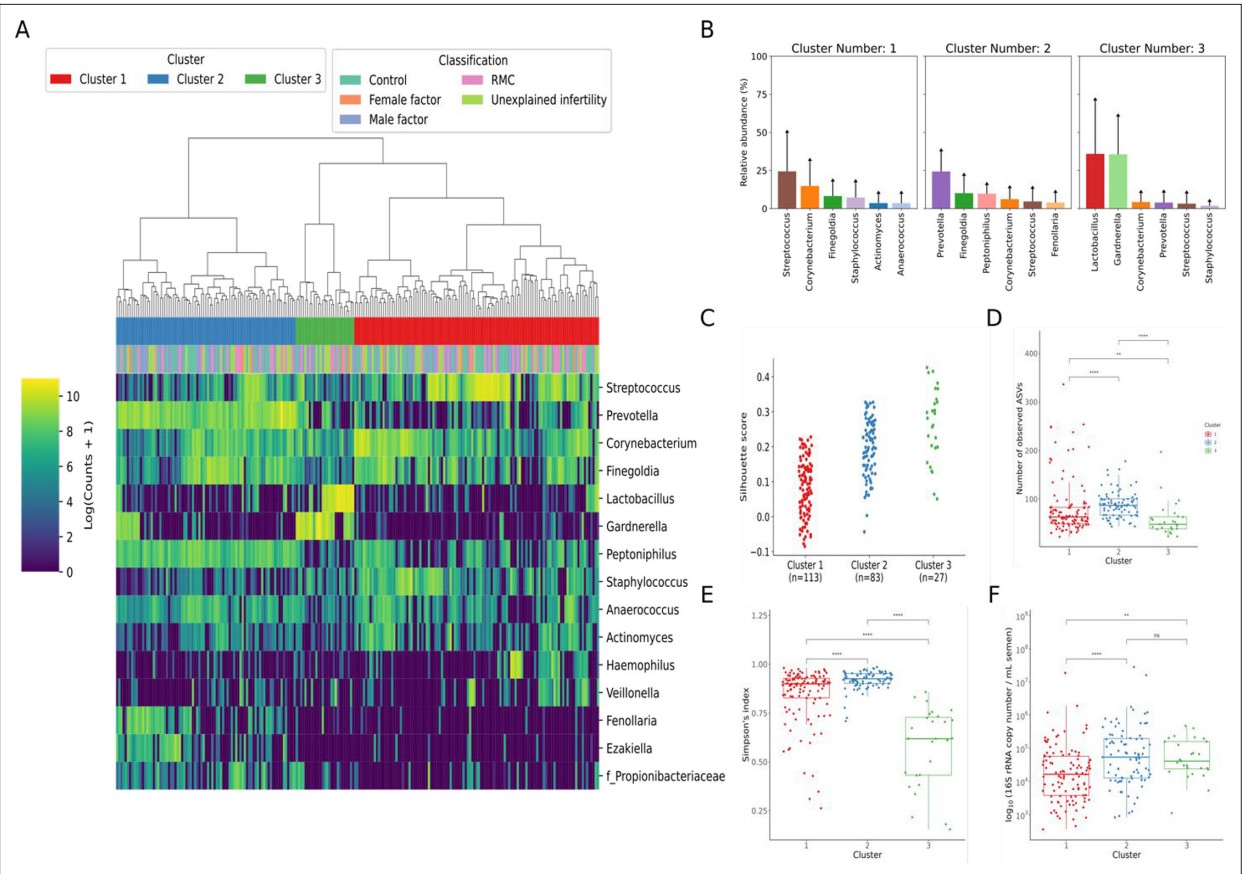

**Figure 1.** Characterisation of semen microbiota composition at genera level. (**A**) Heatmap of Log10 transformed read counts of top 10 most abundant genera identified in semen samples. Samples clustered into three major microbiota groups based mainly on dominance by *Streptococcus* (Cluster 1), *Prevotella* (Cluster 2), or *Lactobacillus* and *Gardnerella* (Cluster 3) (n=223, Ward's linkage). (**B**) Relative abundance of the top 6 most abundant genera within each cluster. (**C**) Silhouette scores of individual samples within each cluster. (**D**) Species richness (p<0.0001; Kruskal-Wallis test) and (**E**) alpha-diversity (p<0.0001; Kruskal-Wallis test) significantly differed across clusters. (**F**) Assessment of bacterial load using qPCR showed Clusters 2 and 3 have significantly higher bacterial loads compared to Cluster 1. Dunn's multiple comparison test was used as a post hoc test for between-group comparisons (*p<0.05, ****p<0.0001).

The online version of this article includes the following figure supplement(s) for figure 1:

**Figure supplement 1.** Genera-level categorisation of seminal microbiota identified three major clusters using average silhouette scores for number of clusters.

clustering analysis at genera level. Taxa belonging to communities 3 and 4 had a high number of connections (higher node degree), both within and between the two communities, including some anti-co-occurrence patterns (SparCC $\rho$ <0). These communities included species from genera *Staphylococcus, Peptoniphilus, Corynebacterium, Prevotella,* among others.

Bacterial richness, diversity, and load were similar between all patient groups examined in the study (*Appendix 2—figure 2*). Similarly, no significant associations between bacterial clusters, richness, diversity, or load with seminal parameters, sperm DNA fragmentation, or semen ROS were observed (*Appendix 1—tables 1 and 2*). No significant differences in relative abundance of bacterial taxa between patient groups were detected, at genus or species level. Several organisms at the genus level, identified variably in the literature as responsible for genitourinary infection, were observed in our dataset, but their prevalence did not reach our criteria (present in at least 25% of the samples) to be carried forward to regression modelling (*Núñez-Calonge et al., 1998*; *Satta et al., 2006*; *Sergerie et al., 2005*). This included *Chlamydia, Ureaplasma, Neisseria, Mycoplasma,* and *Escherichia*. However, several associations (p<0.05) between relative abundance of specific bacterial genera and key sperm parameters were observed (*Table 2*). These included increased sperm DNA fragmentation, which was positively associated with increased relative abundance of *Porphyromonas* and *Varibaculum* and

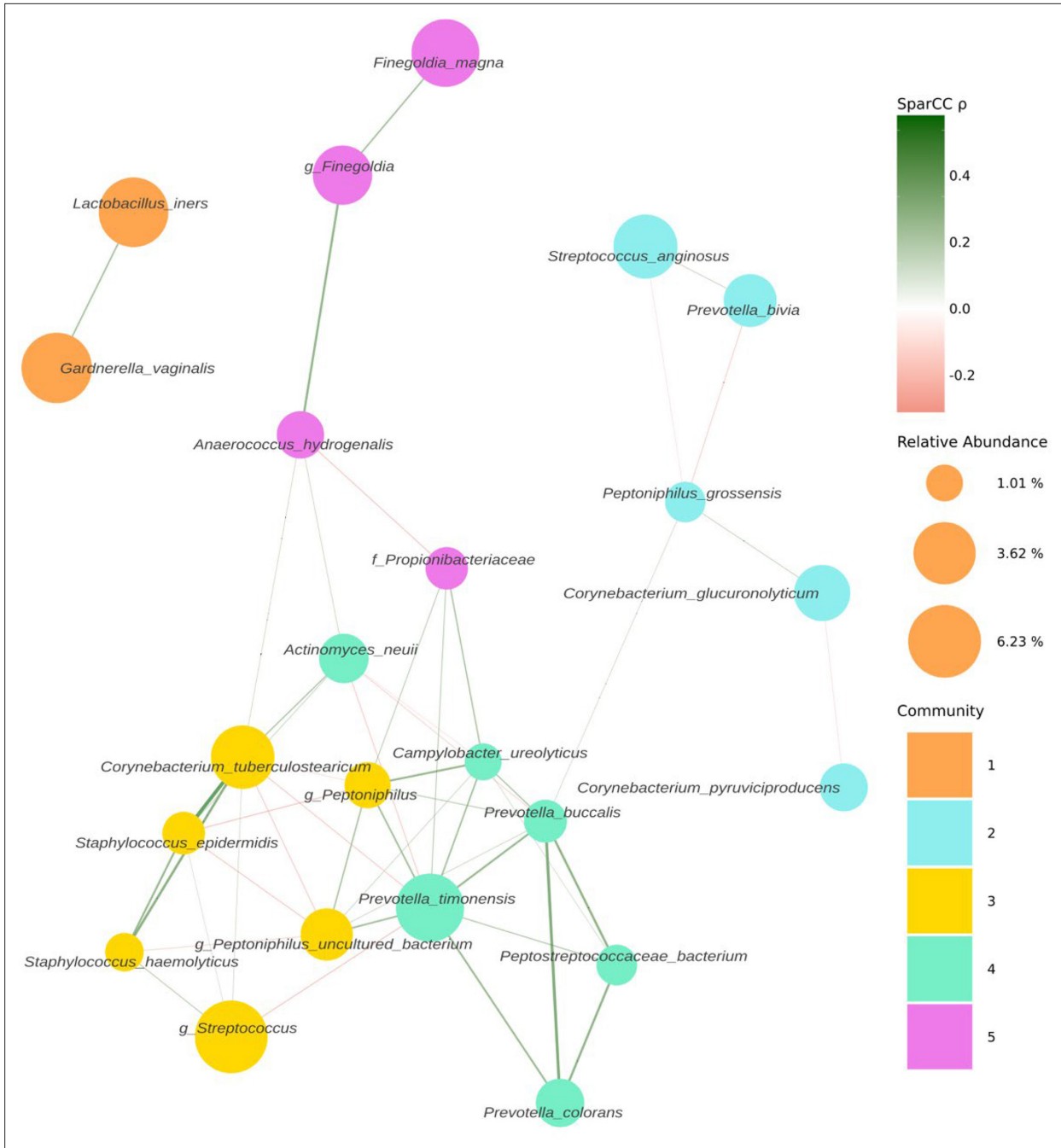

**Figure 2.** Co-occurrence network estimated with SparCC from 16S sequencing counts at species level. Network representing co-occurrence patterns (edges), between various taxonomic units, assigned at species level (nodes). Edges are coloured by their estimated SparCC correlation coefficient ($\rho$). Edges with a SparCC bootstrapped p-value<0.05, $\rho$ <0.25, and singleton nodes are not shown. Node colour represents network community membership. Node sizes are proportional to the mean relative abundance of their respective taxon.

The online version of this article includes the following figure supplement(s) for figure 2:

**Figure supplement 1.** Characterisation of semen microbiota composition at species level.

inversely correlated with *Cutibacterium* and *Finegoldia*. ROS was positively associated with *Lactobacillus* species relative abundance, with analyses performed at species-level taxonomy, indicating that this relationship was largely driven by *L. iners* (p=0.04; *Table 3*). In contrast, *Corynebacterium* was inversely associated with ROS and positively associated with semen volume. Of note, *Flavobacterium* genus was positively associated with both abnormal semen quality and sperm morphology and, in

**Table 2.** Differential abundance analysis for bacterial genera with seminal quality and functional parameters.
Positive t-values indicate a positive relationship, and a negative t-value describes a negative relationship between relative abundance of taxa and seminal quality and function parameters. Significant relationships are indicated using p-values. q-Values represent Benjamini-Hochberg false discovery rate corrected p-values for multiple comparisons.

| Sperm quality and function parameters | Genera | Welch's t-statistic | p-Value | q-Value |
|---|---|---|---|---|
| | *Finegoldia* | –2.36 | 0.01* | 0.27 |
| | *Cutibacterium* | –2.20 | 0.02* | 0.27 |
| | *Porphyromonas* | 2.16 | 0.03* | 0.27 |
| Sperm DNA fragmentation | *Varibaculum* | 2.11 | 0.03* | 0.27 |
| | *Lactobacillus* | 2.18 | 0.02* | 0.66 |
| ROS | *Corynebacterium* | –2.04 | 0.04* | 0.66 |
| | *Flavobacterium* | 3.39 | 0.0008*** | 0.02* |
| Semen quality | *Prevotella* | 2.26 | 0.02* | 0.38 |
| Sperm concentration | *Porphyromonas* | –2.08 | 0.03* | 0.61 |
| | *Flavobacterium* | 3.64 | 0.0003*** | 0.01* |
| Sperm morphology | *Prevotella* | 2.03 | 0.04* | 0.67 |
| | *Corynebacterium* | 2.27 | 0.02* | 0.32 |
| | *Actinotigum* | –2.20 | 0.02* | 0.32 |
| Semen volume | *Varibaculum* | –2.16 | 0.03* | 0.32 |

both cases, withstood FDR correction for multiple testing (q=0.02 and q=0.01, respectively) (**Table 2**; **Figure 3**). Consistent with this, a positive association between an unidentified species of *Flavobacterium* and semen quality was also observed (q=0.01, **Table 3**).

To focus analyses towards the most extreme phenotype of poor semen quality, a sub-analysis of controls compared with MFI was performed (**Table 4**). Non-parametric differential abundance analysis again identified a robust relationship between *Flavobacterium* and abnormal sperm morphology (q=0.01, **Table 4**). At species level, this was mapped to an unidentified species of *Flavobacterium*

**Table 3.** Differential abundance analysis for bacterial species with seminal quality and functional parameters.
Positive t-values indicate a positive relationship, and a negative t-value describes a negative relationship between relative abundance of taxa and seminal quality and function parameters. Significant relationships are indicated using p-values. q-Values represent Benjamini-Hochberg false discovery rate corrected p-values for multiple comparisons.

| Clinical factor | Species | Welch's t-statistic | p-Value | q-Value |
|---|---|---|---|---|
| Sperm DNA fragmentation | *Peptostreptococcaceae bacterium* | 2.18 | 0.03* | 0.91 |
| | *Lactobacillus iners* | 2.24 | 0.02* | 0.94 |
| ROS | Unidentified *Anaerococcus* | –2.03 | 0.04* | 0.94 |
| | Unidentified *Flavobacterium* | 3.76 | 0.0002*** | 0.01* |
| Semen quality | *Corynebacterium tuberculostearicum* | –2.06 | 0.04* | 0.82 |
| | *Corynebacterium tuberculostearicum* | 2.64 | 0.008 | 0.24 |
| | Unidentified *Varibaculum* | –2.48 | 0.01 | 0.24 |
| | *Staphylococcus epidermidis* | 2.35 | 0.01 | 0.24 |
| | Unidentified *Peptoniphilus* | –2.32 | 0.02 | 0.24 |
| | *Dialister propionicifaciens* | –2.24 | 0.02 | 0.24 |
| Semen volume | *Prevotella colorans* | –2.14 | 0.03 | 0.26 |
| Cohorts | *Staphylococcus haemolyticus* | 0.04 | 0.02 | 0.97 |

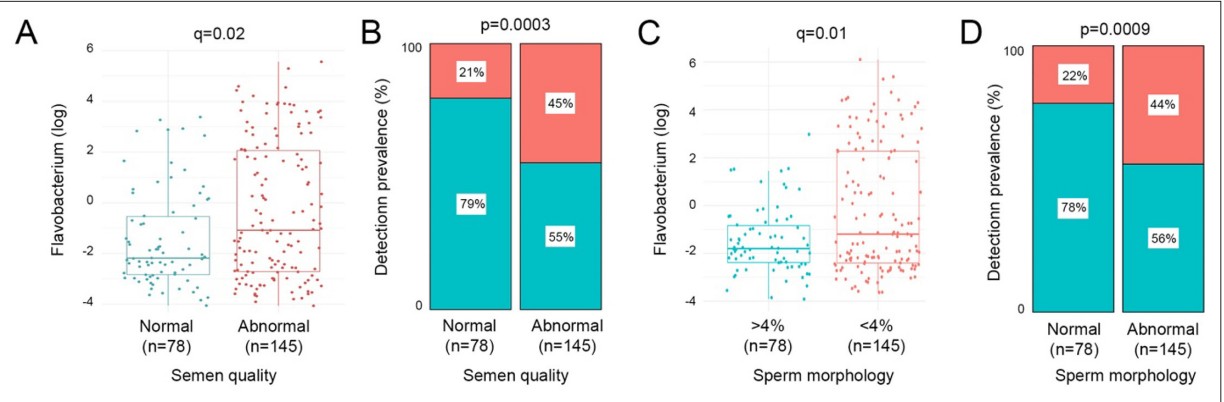

**Figure 3.** Relative abundance and prevalence matrices of *Flavobacterium* in relation to semen quality and morphology. (**A**) Relative abundance of *Flavobacterium* was significantly higher in samples with abnormal semen (p=0.0002, q=0.02). (**B**) Detection of *Flavobacterium* was significantly more prevalent in abnormal semen quality samples (p=0.0003). (**C**) *Flavobacterium* relative abundance was significantly higher in samples with <4% morphologically normal forms (p=0.0002, q=0.01). (**D**) *Flavobacterium* was also significantly more prevalent in samples with low percentage of morphologically normal sperm (p=0.0009).

(q=0.01, *Table 5*). Similar to findings observed for all samples, sperm DNA fragmentation was inversely associated with relative abundance of *Cutibacterium* and positively associated with *Porphyromonas*, and *Varibaculum* was also observed.

## Discussion

We report the largest study to date investigating the seminal microbiota from patients suffering a range of adverse reproductive outcomes, including UI, MFI, and RPL. Moreover, we provide a detailed assessment of the relationship between semen microbial diversity, load, and compositional structure with both molecular and classical seminal parameters. We identified three main clusters present in all study groups. Whilst overall bacterial composition was not associated with aberrations in semen analysis, ROS, and DNA fragmentation, men with unidentified *Flavobacterium* species were more likely to have abnormal semen quality or sperm morphology.

**Table 4.** Differential abundance analysis for specific taxa at *genera* level for controls and cases with male factor infertility.

Positive t-values indicate a relationship, and a negative t-value describes a negative relationship between relative abundance of taxa and seminal quality and function parameters. Significant relationships are indicated using p-values. q-Values represent Benjamini-Hochberg false discovery rate corrected p-values for multiple comparisons.

| Clinical factor | Genera | Welch's t-statistic | p-Value | q-Value |
|---|---|---|---|---|
| Sperm DNA fragmentation | *Cutibacterium* | –2.56 | 0.01* | 0.31 |
| | *Porphyromonas* | 2.34 | 0.02* | 0.31 |
| | *Varibaculum* | 1.96 | 0.051 | 0.53 |
| ROS | *Finegoldia* | –1.99 | 0.04* | 0.77 |
| Sperm concentration | *Finegoldia* | 2.04 | 0.04* | 0.71 |
| Sperm morphology | *Flavobacterium* | 3.64 | 0.0003*** | 0.01* |
| | *Prevotella* | 2.03 | 0.04* | 0.67 |
| Semen volume | *Facklamia* | 2.99 | 0.003** | 0.10 |
| | *Actinotignum* | –2.20 | 0.02* | 0.36 |
| | *Dialister* | –1.99 | 0.04* | 0.36 |

**Table 5.** Differential abundance analysis for specific taxa at species for controls and male factor infertility.

Positive t-values indicate a positive relationship, and a negative t-value describes a negative relationship between relative abundance of taxa and seminal quality and function parameters. Significant relationships are indicated using p-values. q-Values represent Benjamini-Hochberg false discovery rate corrected p-values for multiple comparisons.

| Clinical factor | Species | Welch's t-statistic | p-Value | q-Value |
|---|---|---|---|---|
| Sperm DNA fragmentation | *Staphylococcus hominis* | –2.32 | 0.02* | 0.68 |
| | Unidentified *Flavobacterium* | 2.42 | 0.01 | 0.54 |
| | Unidentified *Anaerococcus* | –2.12 | 0.03 | 0.54 |
| | *Schaalia radingae* | –2.12 | 0.03* | 0.54 |
| ROS | *Haemophilus parainfluenza* | 2.02 | 0.04* | 0.54 |
| Semen quality | Unidentified *Flavobacterium* | 2.36 | 0.01* | 0.91 |
| | *Dialister micraerophilus* | –2.66 | 0.008** | 0.41 |
| | *Corynebacterium tuberculostearicum* | 2.27 | 0.02* | 0.44 |
| | *Staphylococcus epidermidis* | 2.22 | 0.02* | 0.44 |
| Semen volume | *Actinotignum schaalii* | –2.00 | 0.04* | 0.45 |
| Cohorts | *Staphylococcus haemolyticus* | 0.04 | 0.01* | 0.68 |

Several recent studies have indicated the existence of a semen microbiota; however, these studies have been limited to small sample sizes and have failed to reach consensus on the compositional structure of the microbiota or its biological relevance, particularly in the context of sperm function and quality (*Hou et al., 2013*; *Weng et al., 2014*; *Baud et al., 2019*; *Lundy et al., 2021*; *Garcia-Segura et al., 2022*; *Yang et al., 2020*; *Veneruso et al., 2023*). By analysing the samples of 233 men with various reproductive disorders, we offer not only a robust assessment of association between the microbiota and classical seminal parameters, but also key functional parameters including seminal ROS and sperm DNA damage. We incorporated stringent negative controls to permit removal of sequences likely originating from extraction kits and reagents known to contaminate low biomass samples such as semen (*Weng et al., 2014*; *Baud et al., 2019*; *Veneruso et al., 2023*). Molina et al. report that 50–70% of detected bacterial reads may be environmental contaminants in a sample from extracted testicular spermatozoa (*Molina et al., 2021*); with the addition of passage along the urethra, it is likely that contamination of ejaculated semen would be much higher.

Mapping of genera-level relative abundance data enabled semen samples to be categorised into three major clusters characterised by differing relative abundance of *Streptococcus*, *Prevotella*, *Lactobacillus*, and *Gardnerella*. Unlike previous studies, we used an objective statistical approach (i.e. silhouette methods) to determine the optimal number of microbial clusters supported by the data. These findings are largely consistent with earlier semen metataxonomic profiling studies, reporting clusters enriched for *Streptococcus*, *Lactobacillus,* and *Prevotella* (*Hou et al., 2013*; *Weng et al., 2014*; *Baud et al., 2019*). Moreover, Baud et al. reported increased bacterial richness in the *Prevotella*-enriched cluster, which we also observed (*Baud et al., 2019*). This may suggest that certain compositional characteristics of seminal microbiota are conserved across populations. However, similar modelling of species-level data failed to identify statistically robust clusters. This contrasts with other niches such as the vagina where reproducible clusters based on species-level metataxonomic profiles have been demonstrated, reflecting mutualistic relationships between specific species and the host, which have co-evolved over long periods of time (*Pruski et al., 2021*; *France et al., 2020*). It is possible, therefore, that our findings indicate that microbiota detected in semen are likely the result of transient colonisation events. Consistent with this, several species known to be commensal to the penile skin including *Streptococcus*, *Corynebacterium,* and *Staphylococcus*, or the female genital tract including *Gardnerella* and *Lactobacillus*, were observed in semen samples (*Byrd et al., 2018*). This is in keeping with data suggesting microbiota transference during sexual intercourse (*Ma, 2022*). It remains possible that a proportion of bacteria detected in semen reflects contamination of the sample acquired during the collection procedure. Studies undertaking assessment of female partner microbiota profiles as well as temporal profiling of semen microbiota would improve understanding of potential dynamic

restructuring of semen microbiota compositions. This has been done in part by Baud et al. by studying the subfertile couple as a unit to establish if there is a 'couple microbiota' (*Baud et al., 2023*). They took samples from 65 couples with a range of pathologies including idiopathic infertility. From each woman, they took vaginal swabs and follicular fluid samples. From each man, they took semen samples and penile swabs. They undertook extensive negative control series and stringent in silico elimination of possible contaminants. They found the male microbiota to be much more diverse than the female, with 90% of female samples being *Lactobacillus*-dominant. Intra-personal male samples, i.e., semen and penile swabs from the same man, bore more similarity to each other than inter-personal samples of the same sample type, i.e., semen *or* penile swab comparisons between men (*Baud et al., 2023*). They identified that the male microbiota had very little impact on the microbiota of the female sexual partner (*Baud et al., 2023*). Lack of information regarding the sexual activity of the enrolled couples limits this study somewhat.

Several previous studies have described semen microbiota composition to genera level, and some have reported associations between specific genera and parameters of semen quality and function (*Hou et al., 2013*; *Weng et al., 2014*; *Baud et al., 2019*; *Lundy et al., 2021*; *Garcia-Segura et al., 2022*; *Yang et al., 2020*; *Veneruso et al., 2023*). However, in many cases, these studies have failed to consider multiple comparisons testing, likely leading to the reporting of spurious associations. We did not observe any significant associations between bacterial clusters, richness, diversity or load with traditional seminal parameters, sperm DNA fragmentation, or semen ROS. This is in contrast with Veneruso et al., who reported that in infertile patients, semen bacterial diversity and richness was decreased, whereas Lundy et al. reported that diversity was increased in infertile patients (*Lundy et al., 2021*; *Veneruso et al., 2023*). Further, Lundy et al. reported *Prevotella* abundance to be inversely associated with sperm concentration; this was not replicated in our study (*Lundy et al., 2021*). There are several possible reasons accounting for the high heterogeneity in results, including differences in methodology used to assess the microbial component of semen as well as differences in study design (*Farahani et al., 2021*). For example, time of sexual abstinence prior to sample production as well as sample processing time often differs between studies, which has been shown to impact microbiological composition of semen (*Yao et al., 2020*).

The only association between bacterial taxa and semen parameters to withstand false detection rate testing for multiple comparisons detected in our study was between *Flavobacterium* and abnormal semen quality and sperm morphology (q=0.02). The *Flavobacterium* genus taxon we identified as significantly associated with abnormal semen quality and sperm morphology was present in 36.28% of the samples, with a mean relative abundance of 1.15% in those samples. This information and the mention of previous findings of *Flavobacterium* in contamination studies have been added to the discussion. *Flavobacterium* are gram-negative physiologically diverse aerobes, some of which are pathogenic (*Waskiewicz A, 2014*). *Flavobacterium* was recently identified as a dominant genus in immature sperm cells retrieved from testicular biopsies of infertile men in a study by *Molina et al., 2021*. However, in contrast to these findings, a recent smaller study investigating semen collected from 14 sperm donors and 42 infertile idiopathic patients reported an association between *Flavobacterium* and increased sperm motility but a negative correlation with sperm DNA fragmentation (*Garcia-Segura et al., 2022*). The genus *Flavobacterium* was defined in 1923 to encompass gram-negative, non-spore-forming rods of yellow pigment (*Holmes and McMeekin, 1923*). The inclusiveness of this definition resulted in a collective of heterogeneous species. By 1984, the genus had been restricted to those that were also non-motile and non-gliding (*Holmes and McMeekin, 1923*). More recently, with an increase in genomic profiling, many species previously considered to be of genus *Flavobacterium* have been reclassified to genera *Chryseobacterium*, *Cytophaga*, and *Weeksella* (*Bernardet et al., 1996*). Increasing numbers of *Flavobacterium* species are being discovered, such as *gondwanense, collinsii, branchiarum, branchiicola, salegens,* and *scophthalmum* (*Dobson et al., 1993*; *Mudarris et al., 1994*; *Zamora et al., 2013*). The allocation of *Flavobacterium aquatile* to this genus remains controversial due to its motility (*Sheu et al., 2013*). *Flavobacterium* species are widely distributed in the environment, including soil, freshwater, and saltwater habitats (*Loch and Faisal, 2015*; *Vela et al., 2007*). There are many reports of pathogenic infections of *Flavobacterium* species in fish; however, human infections are rare (*Zamora et al., 2013*). A handful of case reports have described opportunistic infections, including pneumonia, urinary tract infection, peritonitis, and meningitis (*Sung et al., 2015*; *Tian et al., 2011*; *Park and Ryoo, 2016*; *Mosayebi et al., 2011*).

*Flavobacterium lindanitolerans* and *Flavobacterium ceti* have been isolated as causative agents in some (*Zurbuchen et al., 2023*; *Park and Ryoo, 2016*). Case reports also describe *Flavobacterium odoratum* as a causative agent in urinary tract infection, most often in the immunocompromised or those with indwelling devices (*Benedetti et al., 2011*; *Chauhan et al., 2020*; *Verma et al., 2018*). However, this was one of many species previously of genus *Flavobacterium* reclassified, in this case to genus *Myroides* (*Vancanneyt et al., 1996*). Notably, in our sample, participants were asymptomatic of urinary tract infection.

Though notwithstanding multiple corrections, we did observe several other associations between specific bacterial taxa and semen parameters. For example, samples enriched with *Lactobacillus* had lower incidence of elevated seminal ROS, a relationship which could largely be accounted for by *L. iners*, a common member of the cervicovaginal niche (*MacIntyre et al., 2017*). Various studies have also found *Lactobacillus* enrichment in semen to associate with normal seminal parameters, especially morphology (*Weng et al., 2014*; *Baud et al., 2019*), where *Lactobacillus* is predominant (*Weng et al., 2014*). However, an association between samples enriched with *Lactobacillus* and asthenospermia or oligoasthenospermia has also been described (*Yang et al., 2020*). We also observed an association between increased sperm DNA fragmentation and samples enriched with *Varibaculum*, which is consistent with previous reports of increased relative abundance of *Varibaculum* in semen of infertile men (*Veneruso et al., 2023*).

A limitation of this and other similar studies is that it was a single institutional study with limited ethnic diversity and potential geographical changes induced by environment or dietary habit. Gut microflora is known to display geographical variability; we cannot exclude that similar geographical variability exists for the seminal microbiota (*Suzuki and Worobey, 2014*). The universal primers used during NGS may not be universal and may anneal variably to specific bacteria, resulting in over-detection, under-detection, or indeed non-detection of some taxa (*Forney et al., 2004*; *Wintzingerode et al., 1997*). A further limitation of this study, and others, is the lack of reciprocal genital tract microbiota testing of the female partners, or paired seminal and urinary samples from male participants. Additionally, we did not have other covariables such as smoking status with which to include in further analyses.

In summary, our study confirms that compositionally, the semen microbiota can be broadly classified into three major groups based upon relative abundance of key bacterial genera. Despite different methodological approaches, a number of studies, including our own, indicate a *Prevotella*-dominated or *Lactobacillus*-dominated seminal microbiota, perhaps suggesting a stable microbiota at genera level. Our species-level data, however, failed to show similar clusters, perhaps instead suggesting transient colonisation. Longitudinal studies are required to ascertain the stability of the seminal microbiota. We provide evidence for an association between *Lactobacillus* abundance and normal seminal parameters. However, our results indicate that no specific semen bacterial composition can robustly differentiate between fertile and infertile men, although a small subset of bacteria may be associated with changes in seminal parameters. Our finding that an unidentified species of *Flavobacterium* impairs seminal parameters warrants further exploration and may offer the potential for targeted therapies. Larger, multicentred studies as well as mechanistic investigations are required to establish causal links between the semen microbiota and male fertility.

## Materials and methods

*Ethical approval* was granted by the West London and Gene Therapy Advisory Committee (GTAC) Research Ethics Committee (approval identifier: 14/LO/1038) and by the Internal Review Board at the Centre for Reproductive and Genetic Health (CRGH) (approval identifier: IRB-0003C07.10.19). Participants were recruited following informed consent from clinics in Imperial College London NHS Trust and the CRGH. No individually identifiable data is presented, so separate consent to publish from participants was not required. Further detailed information on methods used in this study is included in the Supplementary Material.

*Semen samples* were produced by means of masturbation after 3–7 days abstinence. All semen samples were collected into sterile containers after cleaning of the penis using a sterile wipe. Samples were incubated at 37°C for a minimum of 20 min prior to analysis. An aliquot was collected in a sterile cryovial and stored at –80°C.

*Diagnostic semen analysis* was carried out according to WHO 2010 guidelines and UK NEQAS accreditation (*World Health Organization, 2021*; *Scheme UNRS, 2023*). Seminal analysis was performed in the Andrology Departments of Hammersmith Hospital and CRGH. Microscopic and macroscopic semen qualities were assessed within 60 min of sample production. Semen volume, sperm concentration, total sperm count, progressive motility and total motility count, morphological assessment, anti-sperm antibodies, and leucocyte count were established.

*ROS analysis* was performed using an in-house developed chemiluminescence assay validated by *Vessey et al., 2014*. Results are therefore reported as 'relative light units per second per million sperm'. The upper limit of optimal ROS was internally determined at 3.77 RLU/s/$10^6$ sperm (95% CI) (*Sergerie et al., 2005*).

*Sperm DNA fragmentation assessment* performed by terminal deoxynucleotidyl transferase biotin-dUTP nick end labelling (TUNEL) assay defined elevated sperm DNA fragmentation as >20% (*Sharma et al., 2021*). Samples for the Comet assay were sent to the Examen Lab (Belfast, UK) for analysis with elevated sperm DNA fragmentation defined as >27% (*Hughes et al., 1996*).

*DNA extraction* was performed on 200 µl of semen using enzymatic lysis and mechanical disruption. Bacterial load was estimated by determining the total number of 16S rRNA gene copies per sample using the BactQuant assay (*Liu et al., 2012*).

*Metataxonomic profiling of semen microbiota* was performed using MiSeq sequencing of bacterial V1-V2 hypervariable regions of 16S rRNA genes using a mixed forward primer set 28F-YM GAGTTTGATYMTGGCTCAG, 28F-Borrelia GAGTTTGATCCTGGCTTAG, 28F-Chloroflex GAATTTGA TCTTGGTTCAG, and 28F-Bifdo GGGTTCGATTCTGGCTCAG at a ratio of 4:1:1:1 with 388R reverse primers. Sequencing was performed on the Illumina MiSeq platform (Illumina, Inc, San Diego, CA, USA). Following primer trimming and assessment of read quality, amplicon sequence variant (ASV) counts per sample were calculated and denoised using the QIIME2 pipeline (*Bolyen et al., 2019*) and the DADA2 algorithm (*Callahan et al., 2016*). ASVs were taxonomically classified to species level using a naive Bayes classifier trained on all sequences from the V1-V2 region of the bacterial 16S rRNA gene present in the SILVA reference database (release 138.1) (*Quast et al., 2013*; *Davis et al., 2018*).

Further methodological details can be found in Appendix 3.

## Controls and contamination

Three negative kit/environmental control swabs were included to identify and eliminate potential sources of contamination and false positives in the 16S *metataxonomic profiles*. These swabs were removed from the manufacturer's packaging, waved in air, and then subjected to the same entire DNA extraction protocol. Decontamination of data was done using the decontam package (v1.9.0) in R, at ASV level, using both 'frequency' and 'prevalence' contaminant identification methods with *threshold* set to 0.1 (*Davis et al., 2018*). The 'frequency' filter was applied using the total 16S rRNA gene copies measured as the *conc* parameter. For the 'prevalence' filter, all three blank swabs were used as negative controls and compared against all semen samples. ASVs classified as a contaminant by either method (n=94) were excluded.

## Statistical analysis

Hierarchical clustering with Ward's linkage and Jensen-Shannon distance was used to assign samples to putative community state types, with the number of clusters chosen to maximise the mean silhouette score. Linear regression models used to regress microbiota features against semen quality parameters and other clinical and demographic variables were fitted with the base R *lm* function (v4.2.0). The Benjamini-Hochberg false discovery rate (FDR) correction was used to control the FDR of each covariate signature independently (e.g. ROS, DNA fragmentation, or semen quality), with a q<0.05, or 5%, cut-off, in both regression and chi-squared analyses. Detailed information for statistical modelling is presented in Appendix 3.

## Acknowledgements

We would like to thank the patients and participants for their involvement in the study. The Section of Endocrinology and Investigative Medicine is funded by grants from the MRC, NIHR, and is supported by the NIHR Biomedical Research Centre Funding Scheme and the NIHR/Imperial Clinical Research Facility. The views expressed are those of the author(s) and not necessarily those of Tommy's, the

NHS, the NIHR, or the Department of Health. The following authors are also funded as follows: NIHR Research Professorship (WSD), NIHR Post-Doctoral Fellowship (CNJ). This project was supported by a research grant from Charm Foundation UK as well as funding by Tommy's National Centre for Miscarriage Research (grant P62774).

## Additional information

### Competing interests
Channa N Jayasena: Receives an investigator-led grant from LogixX Pharma Ltd. The other authors declare that no competing interests exist.

### Funding

| Funder | Grant reference number | Author |
|---|---|---|
| Charm Foundation UK | | Channa N Jayasena |
| Tommy's National Centre for Miscarriage Research | P62774 | David A MacIntyre |
| Logixx Pharma Ltd | | Channa N Jayasena |

The funders had no role in study design, data collection and interpretation, or the decision to submit the work for publication.

### Author contributions
Shahriar Mowla, Data curation, Formal analysis, Investigation, Methodology, Project administration, Writing – review and editing; Linda Farahani, Conceptualization, Data curation, Formal analysis, Investigation, Methodology, Project administration, Writing – review and editing; Tharu Tharakan, Data curation, Formal analysis, Investigation, Methodology, Project administration; Rhianna Davies, Formal analysis, Writing – original draft; Goncalo DS Correia, Data curation, Formal analysis, Writing – review and editing; Yun S Lee, Writing – review and editing; Samit Kundu, Shirin Khanjani, Jara Ben Nagi, Phillip Bennett, Conceptualization, Writing – review and editing; Emad Sindi, Investigation, Project administration; Raj Rai, Conceptualization, Supervision, Investigation, Writing – review and editing; Lesley Regan, Conceptualization, Investigation, Writing – review and editing; Dalia Khalifa, Data curation, Investigation, Project administration; Ralf Henkel, Conceptualization, Methodology; Suks Minhas, Supervision, Writing – review and editing; Waljit S Dhillo, Conceptualization, Supervision, Writing – review and editing; David A MacIntyre, Conceptualization, Formal analysis, Supervision, Funding acquisition, Investigation, Visualization, Methodology, Writing – original draft, Project administration, Writing – review and editing; Channa N Jayasena, Conceptualization, Data curation, Supervision, Funding acquisition, Methodology, Writing – original draft, Writing – review and editing

### Author ORCIDs
Rhianna Davies ⓘ https://orcid.org/0000-0001-6922-4553
Goncalo DS Correia ⓘ https://orcid.org/0000-0001-8271-9294
David A MacIntyre ⓘ https://orcid.org/0000-0002-4186-5567
Channa N Jayasena ⓘ https://orcid.org/0000-0002-2578-8223

### Ethics
Ethical approval was granted by the West London and Gene Therapy Advisory Committee (GTAC) Research Ethics Committee (approval identifier: 14/LO/1038) and by the Internal Review Board at the Centre for Reproductive and Genetic Health (CRGH) (approval identifier: IRB-0003C07.10.19). Participants were recruited following informed consent from clinics in Imperial College London NHS Trust and The Centre for Reproductive and Genetic Health (CRGH). No individually identifiable data is presented, so separate consent to publish from participants was not required.

Reviewer #1 (Public review): https://doi.org/10.7554/eLife.96090.4.sa1
Author response https://doi.org/10.7554/eLife.96090.4.sa2

# Additional files

## Supplementary files
MDAR checklist

## Data availability
Data and material availability statement: The 16S rRNA metataxonomic dataset and the data analysis scripts are publicly available at the European Nucleotide Archive (project accession PRJEB57401) and GitHub (repository link https://github.com/Gscorreia89/semen-microbiota-infertility, copy archived at Zenodo: *Correia, 2025*) respectively.

The following dataset was generated:

| Author(s) | Year | Dataset title | Dataset URL | Database and Identifier |
|---|---|---|---|---|
| Mowla S, Farahani L, Tharakan T | 2024 | Characterisation and comparison of semen microbiota and bacterial load in men with infertility, recurrent miscarriage, or proven fertility | https://www.ebi.ac.uk/ena/browser/view/PRJEB57401 | EBI European Nucleotide Archive, PRJEB57401 |

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

# Appendix 1

**Appendix 1—table 1.** Comparison of mean age and prevalence of ethnicities in study recruitment cohorts.

Ethnicity representation amongst recruited cohorts was not significantly different (p=0.38, chi-squared test). RPL: recurrent pregnancy loss, MFI: male factor infertility, UI: unexplained infertility.

| Study cohort | Age (mean ±SD) | Ethnicity |
|---|---|---|
| Control (n=63) | 40.1±8 | 39/63 (62%) Caucasian |
| | | 24/63 (38%) Non-Caucasian |
| RPL (n=46) | 38.2±5 | 35/46 (76%) Caucasian |
| | | 11/46 (24%) Non-Caucasian |
| MFI (n=58) | 36.3±4.5 | 41/58 (70%) Caucasian |
| | | 17/58 (30%) Non-Caucasian |
| UI (n=56) | 37±4.7 | 41/56 (73%) Caucasian |
| | | 15/56 (27%) Non-Caucasian |

**Appendix 1—table 2.** Distribution of clinical factors, microscopic seminal parameters, confounding factors, and recruitment cohorts according to genera clusters.

Chi-squared tests.

| Factors | Thresholds | Cluster 1 | Cluster 2 | Cluster 3 | p-Value |
|---|---|---|---|---|---|
| DNA frag index | Low | 60 (53%) | 39 (34%) | 15 (13%) | |
| | High | 37 (45%) | 35 (43%) | 10 (12%) | 0.47 |
| ROS | <3.77 RLU/s | 74 (52%) | 56 (39%) | 13 (9%) | |
| | >3.77 RLU/s | 19 (58%) | 11 (33%) | 3 (9%) | 0.81 |
| Semen volume | Optimal | 105 (50%) | 80 (38%) | 23 (12%) | |
| | Suboptimal | 8 (53%) | 3 (20%) | 4 (27%) | 0.12 |
| Cohorts | Control | 36 (57%) | 22 (35%) | 5 (8%) | |
| | MFI | 26 (45%) | 25 (45%) | 7 (10%) | |
| | RPL | 23 (50%) | 17 (37%) | 6 (13%) | |
| | UI | 28 (50%) | 19 (34%) | 9 (16%) | 0.76 |
| Age | <34 | 30 (61%) | 14 (29%) | 5 (10%) | |
| | 34–41 | 59 (48%) | 49 (40%) | 16 (12%) | |
| | >41 | 24 (48%) | 20 (40%) | 6 (12%) | 0.58 |
| Ethnicity | Caucasian | 82 (53%) | 57 (37%) | 17 (10%) | |
| | Non-Caucasian | 31 (46%) | 26 (39%) | 10 (15%) | 0.58 |
| Concentration | >15 M/ml | 93 (51%) | 67 (37%) | 22 (12%) | |
| | <15 M/ml | 20 (49%) | 16 (39%) | 5 (12%) | 0.96 |
| Progressive motility | >32% | 105 (51%) | 78 (38%) | 24 (11%) | |
| | <32% | 8 (50%) | 5 (31%) | 3 (19%) | 0.67 |
| Morphology | >4% | 37 (50%) | 24 (32%) | 13 (18%) | |
| | <4% | 72 (50%) | 58 (40%) | 14 (10%) | 0.19 |
| Semen quality | Optimal | 41 (53%) | 24 (31%) | 13 (16%) | |
| | Suboptimal | 72 (50%) | 59 (17%) | 14 (33%) | 0.17 |

**Appendix 1—table 3.** Richness and diversity of seminal bacterial based on seminal quality and function parameters.

Categorical classifications of seminal parameters were based on the clinically defined thresholds. Mann-Whitney tests for all except age. Kruskal-Wallis test was used for age.

| Factors | Richness p-value | Diversity p-value |
| --- | --- | --- |
| DNA frag index | 0.68 | 0.89 |
| ROS | 0.25 | 0.23 |
| Semen volume | 0.54 | 0.85 |
| Age | 0.14 | 0.12 |
| Ethnicity | 0.31 | 0.24 |
| Concentration | 0.79 | 0.66 |
| Progressive motility | 0.38 | 0.54 |
| Morphology | 0.82 | 0.97 |
| Semen quality | 0.74 | 0.90 |

## Appendix 2

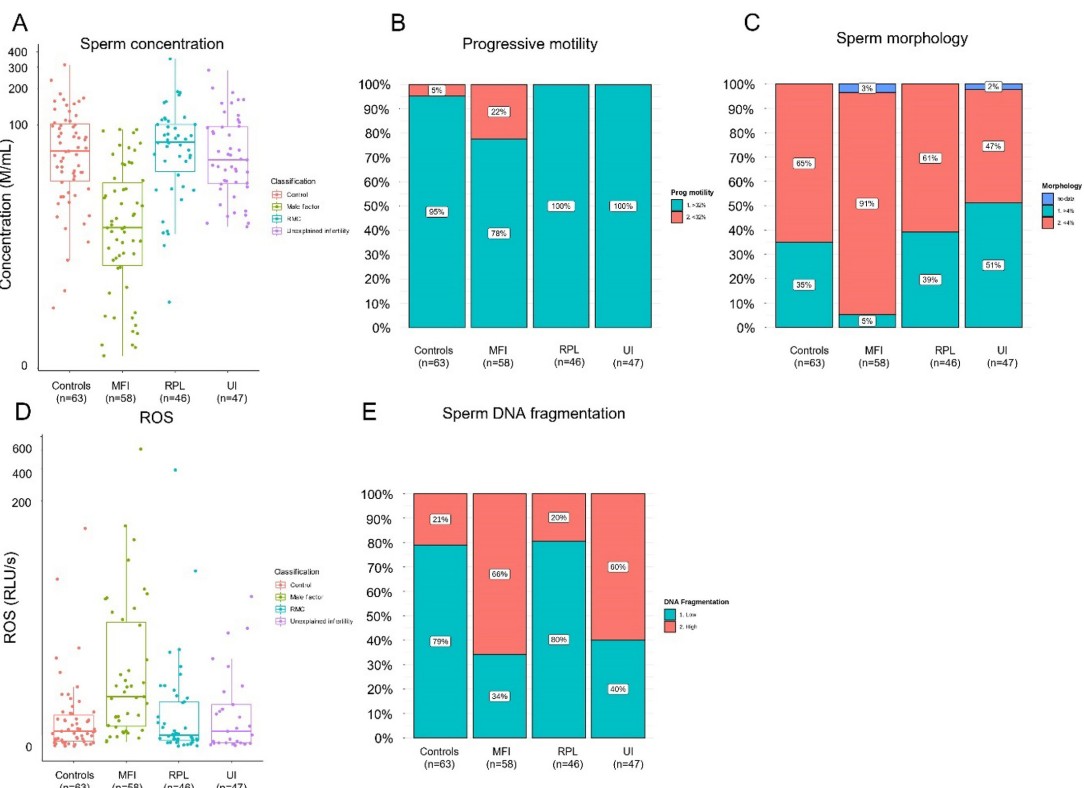

**Appendix 2—figure 1.** Seminal quality and function parameters according to recruited cohorts. Comparison of microscopic semen parameters (**A**) concentration (p<0.0001, Kruskal-Wallis rank-sum test), (**B**) progressive motility (p<0.0001, Pearson's chi-squared test), and (**C**) morphology (p<0.0001, Pearson's chi-squared test) suggested poor semen quality for male factor infertility (MFI) patients. Comparison of clinical semen qualities: (**D**) reactive oxidative species (ROS), (**E**) sperm DNA fragmentation index.

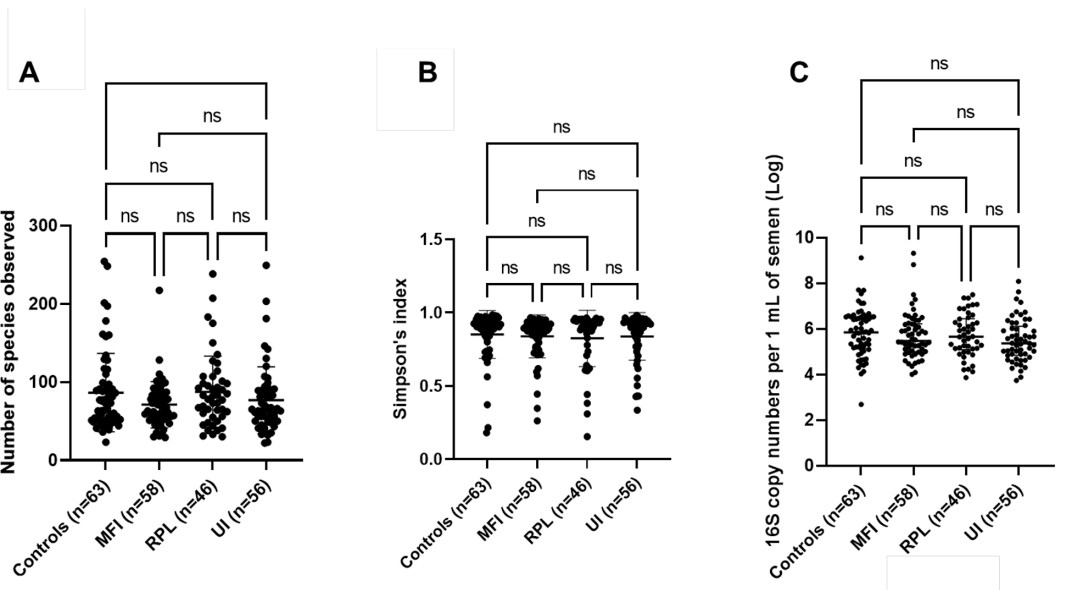

**Appendix 2—figure 2.** Ecological parameters of seminal microbiota for the recruited study cohorts. (**A**) Species richness (p=0.30) and (**B**) Simpson's diversity index (p=0.49) were not significantly different based on recruited

*Appendix 2—figure 2 continued*

study cohorts. Kruskal-Wallis tests with Dunn's multiple comparison p-values demonstrated on the plots. (**C**) Bacterial load of seminal microbiota in recruited study cohorts. There were no significant differences in bacterial load based on recruited study cohorts using the number of 16S rRNA genes per 1 ml of semen (p=0.22, Kruskal-Wallis test).

## Appendix 3

### Methods supplement

#### Study design and patient recruitment

This was a case-control clinical study, the experimental arm included men with MFI, UI, and male partners of women with RPL. Ethical guidelines for human research were adhered to, and all participants provided written informed consent. Ethical approval was granted by the West London and GTAC Research Ethics Committee (14/LO/1038) and by the Internal Review Board at CRGH (IRB-0003C07.10.19).

Samples were recruited between January 2019 and March 2020. Participants were recruited from The Centre for Reproductive & Genetic Health (CRGH, 230–232 Great Portland Street, London, W1W 5QS), the recurrent miscarriage clinic at St. Mary's Hospital (Praed Street, London W2 1NY). Participants in the control group were recruited through local posters at two hospital sites: St. Mary's Hospital and Hammersmith Hospital, London (72 Du Cane Road, London W12 0HS). Inclusion criteria for both groups were men aged 18–60 years of age and BMI <35 kg/m$^2$. To be included in the study group, participants must have had at least two consecutive miscarriages, and to be included in the control group, the participants must have a history of conceiving and having a child with their partner. Exclusion criteria were a confirmed female cause for RPL, current symptoms of genitourinary tract infections, alcohol excess, hormone therapy, smoking within the last 6 months, and active treatment for severe systemic disease. Once enrolled, no sample or data point was excluded once obtained.

#### Sample collection

Semen samples were produced by means of masturbation. Patients were asked to wash and dry their hands thoroughly and capture the entire sample in the sterile pots provided. The pots were labelled with patient name, date of birth, and other relevant information. Upon completion, they then placed the collection pot inside a hatch from which the laboratory staff collected it. It was then placed in an incubator at 37°C for a minimum of 20 min prior to analysis and taking an aliquot for storage. The aliquots were collected in sterile cryovials and stored at –80°C until the day of analysis.

#### Semen analysis

Semen analysis was carried out by qualified Andrologists in the Andrology Departments of both Hammersmith Hospital and CRGH according to WHO 2010 guidelines and UK NEQAS accreditation (*World Health Organization, 2021*). Microscopic and macroscopic semen qualities were assessed within 60 min of sample production. This included measurement of semen volume, sperm concentration, total sperm count, progressive motility and total motility count, morphological assessment, anti-sperm antibodies, and leucocyte count. Measurements were performed once for each sample.

#### ROS analysis

ROS levels in semen samples were analysed in the Andrology laboratory using an in-house-developed chemiluminescence assay validated by *Vessey et al., 2014*. In this assay, luminol is oxidised in the presence of ROS, which generates chemiluminescence. ROS is measured indirectly by measuring the chemiluminescence generated using a CE-marked luminometer (GloMax Promega Corporation) (*Vessey et al., 2014*). In advance of ROS measurement, a 100 mmol/l luminol (5-amino-2,3-dihydro-1,4-phthalazinedione; Sigma-Aldrich) stock solution was prepared in dimethylsulfoxide (DMSO). The stock solution generally remains stable for up to 15 weeks. Due to light sensitivity of the assay, the analysis was done in the dark and in an aluminium foil-covered polystyrene Falcon tube. All reagents stored at 4°C were brought to room temperature before analysis. A luminol working solution (5 mmol/l luminol prepared in DMSO), negative controls (400 µl PBS with 10 µl 5 mmol/l luminol working solution), and positive control samples (395 µl PBS, 5 µl 30% H$_2$O$_2$ (VWR), and 10 µl 5 mmol/l luminol working solution). On the morning of each analysis, the luminometer was calibrated using the negative and positive controls. All samples were mixed gently immediately before placing the tubes in the luminometer. Sperm concentration of >1 M/ml is required for this assay. This is because ROS measurement could be unreliable in lower than 1 M/ml sperm concentrations (*Agarwal et al., 2004*). A total of 400 µl of semen was aliquoted using a positive displacement pipette and placed into a 1.5 ml microfuge tube. Following this, 10 µl of the luminol working solution was added and gently mixed before taking readings in the luminometer. This process is time sensitive and was done

at exactly 20 min of sample production (minimum amount of time for semen to reach liquefaction). A total of 10 consecutive readings were recorded at 1 min apart. The ROS value was obtained by subtracting the control mean from the test mean and correcting it for sperm concentration. Results are therefore reported as 'relative light units per second per million sperm'. The upper limit of optimal ROS was internally determined at 3.77 RLU/s/10$^6$ sperm (95% CI). Measurements were performed once for each sample.

## Sperm DNA fragmentation assessment

For 183 samples TUNEL assay (n=183) was used locally. Elevated sperm DNA fragmentation was defined as >20% via TUNEL assay (*Sharma et al., 2021*). For the remaining 40 samples, Comet assay (a single gel electrophoresis assay) was used. Samples for the Comet assay were sent to the Examen Lab (Belfast, UK) for analysis. Elevated sperm DNA fragmentation was defined as >27% via Comet assay (*Hughes et al., 1996*). Measurement was performed once for each sample.

## Controls and contamination

A series of negative kit/environmental controls were included to identify potential sources of contaminant and elimination of contaminants. Identification and removal of contaminant ASVs was done using the decontam package (v1.9.0) in R (*Davis et al., 2018*).

## DNA extraction

DNA extraction was performed using the QIAamp DNA Mini Kit as per the manufacturer's instructions (*QIAGEN, 2016*). Samples were removed from –80°C storage on the day of the analysis and placed on wet ice to thaw. 200 µl of semen was used for the process of DNA extraction. Samples were collected in an ultraviolet-radiated 2 ml Eppendorf tube. 300 µl of enzymatic master mix, including lysozyme 10 mg/ml, mutanolysin 25 U/µl, lysostaphin 4000 U/ml, was added. After the addition of the enzymatic master mix, the samples were pulse-vortexed to mix the pellet with the enzymes and incubated at 37°C for 1 hr. Following the incubation period, samples were pulse-centrifuged to bring down any condensation gathered on the inside of the lid or on the sides of the tubes. After completion of the enzymatic lysis, the cells in the solution were further disrupted by bead beating. A total of 100 mg of bleached and rinsed 0.1 mm diameter zirconia/silica was added to each tube and oscillated at 25 Hz for 1 min using a TissueLyser. The samples were then pulse-spun, and 200 µl of the solution was transferred to a sterile 1.5 ml tube with care taken not to carry forward any beads into the new tube. A total of 20 µl of proteinase K was added to each tube along with 200 µl of buffer AL and mixed. The samples were then incubated at 56°C for 30 min before adding 200 µl of absolute ethanol and pulse vortexing and pulse centrifugation. The solution was then added to a QIAamp (*QIAGEN, 2016*) mini spin column in a 2 ml collection tube without wetting the rim of the column. The caps were then firmly shut, and the tubes were centrifuged at 6200 × *g* for 1 min. The collection tubes were then discarded, and the columns were placed in fresh 2 ml collection tubes. Next, 500 µl of AW1 buffer was added without wetting the column rim, and the tubes were centrifuged at 6200 × *g* for 1 min. Columns were then placed in a fresh collection tube, and 500 µl of AW2 buffer was added. Column lids were firmly shut, and the tubes were centrifuged at 17,000 × *g* for 3 min. The columns were placed in a fresh collection tube and centrifuged at 17,000 × *g* for 1 min to eliminate the chance of possible AW2 buffer carryover.

The extraction continued by placing the columns in a sterile 1.5 µl tube, and 100 µl of AE buffer was added. The solutions were then incubated at room temperature for 10 min and further centrifuged at 17,000 × *g* for 1 min. 5 µl of the eluates were aliquoted for PCR to confirm successful extraction. 20 µl of the eluates were separately aliquoted, stored at –20°C, and sent for sequencing to Research and Testing Lab in Texas, USA, using Illumina MiSeq platform (Illumina, Inc, San Diego, CA, USA).

## Quantitative bacteriology

Bacterial load was estimated by determining the total number of 16S rRNA gene copies per sample (*Liu et al., 2012*). A known concentration of *Escherichia coli* DNA was used to create a tenfold standard curve from 300 to 30,000,000 copies of 16S DNA. The *E. coli* standards and the 5 µl of sample DNA templates were combined with a platinum PCR supermix UDG containing 50 nM Rox, BactQuant forward primer at 100 µM (5' CCT ACG GGA GGC AGC A) and reverse primer at 100 µM (: 5' GGA CTA CCG GGT ATC TAA TC), and probe (5' 6FAM-CAG CAG CCG CGG TA-MGBNFQ) were loaded in duplicates onto a PCR plate. Following a 2 min centrifugation, the plate was transferred to the StepOne qPCR machine. PCR parameters were set to hold at 50°C for 2 min, 95°C for 10 s,

followed by 40 cycles of 95°C for 15 s and 60°C for 1 min before a final holding stage of 4°C until plate retrieval. StepOne software was used to calculate the 16S DNA copy number of the sample in template in relation to the *E. coli* standard curve. Template-free PCR controls were included in each run to exclude contamination of PCR reagents.

## 16S rRNA gene sequence processing

Mixed forward primers 28F-YM GAGTTTGATYMTGGCTCAG, 28F-Borrellia GAGTTTGATCCTGGCT TAG, 28F-Chloroflex GAATTTGATCTTGGTTCAG, and 28F-Bifdo GGGTTCGATTCTGGCTCAG at a ratio of 4:1:1:1 with 388R reverse primers were used to amplify the V1-V2 hypervariable regions of 16S rRNA gene amplicons. Sequencing was performed on the Illumina MiSeq platform (Illumina, Inc, San Diego, CA, USA). Primer sequences were trimmed using cutadapt, and read quality was checked using FastQC (*Martin, 2011*; *Babraham Bioinformatics, 2010*). ASV counts per sample were calculated and denoised using the QIIME2 pipeline (*Bolyen et al., 2019*) and the DADA2 denoising algorithm (*Callahan et al., 2016*). ASVs were taxonomically classified to species level using a naive Bayes classifier trained on all sequences from the V1-V2 region of the bacterial 16S rRNA gene present in the SILVA reference database (release 138.1) (*Quast et al., 2013*; *Bokulich et al., 2018*). Extraction kit and negative reagent controls were used to identify and exclude potential contaminants using the decontam package (v1.9.0) in R (*Davis et al., 2018*).

## Statistical analysis

Hierarchical clustering with Ward's linkage and Jensen-Shannon distance was used to assign samples to putative community state types, with the number of clusters chosen to maximise the mean silhouette score. Hierarchical clustering analyses and associated figures were performed in Python (v3.9.12) and with the '*pandas*' (v1.4.2), '*scipy*' (v1.7.3), '*matplotlib*' (v3.5.1), and '*seaborn*' (v0.11.2) libraries. Principal coordinate analyses with Jensen-Shannon distance were implemented in R (v4.2.0) using the packages '*vegdist*' (v1.40.0) and '*ecodist*' (v2.0.9), and figures generated with '*ggplot2*' (v3.3.6). Network co-occurrence analyses were also performed in R, with the '*SpiecEasi*' (v1.1.3) implementation of the SparCC method. Species-level count data were filtered by excluding all taxa with a mean relative abundance lower than 1%. SparCC correlations were bootstrapped (10,000 samples), and pairwise correlations with a bootstrapped p-value<0.05, or SparCC $\rho$ <0.25, were removed. Network visualisation and community detection (using the Louvain method) was performed with the '*igraph*' (v2.0.3), '*qgraph*' (v1.9.5), '*ggraph*' (v2.1.0), and '*tidygraph*' (v1.2.3) (*Blondel et al., 2008*). Linear regression models used to regress microbiome features against semen quality parameters and other clinical and demographic variables were fitted with the base R *lm* function. Only features present (non-zero counts) in at least 25% of the samples were carried forward for regression modelling. Centred-log-ratio (clr) transformed (with the '*propr*' [v4.2.6] package) species or genera count values were modelled using linear models with the following formula: *clr*(Species)~covariate. For the analyses with estimated absolute species concentrations, log transformation was used instead of the clr transform: *log*(Species + 1)~covariate. The '*emmeans*' package (v1.7.4.1) was used to perform contrast coding and obtain respective effect size estimates and *Welch's t-test p-values*. Chi-squared tests were calculated with the base R *chisq.test* function with Monte Carlo simulation of p-values and 1000 resampling replicates (*simulate.p.value=T, B=1000*). The Benjamini-Hochberg FDR correction (via the *p.adjust* R base function) was used to control the FDR of each covariate signature independently (e.g. ROS, DNA fragmentation, or semen quality), with a q<0.05, or 5%, cut-off, in both the regression and chi-squared analyses. The '*ggplot2*' (v3.3.6), '*ggstatsplots*' (v0.9.4), and '*scales*' (v1.20) packages were used for figure generation.

