## [Editor Report · eLife Assessment]

This **valuable** study reports a potential connection between the seminal microbiome and sperm quality/male fertility. The data are generally **convincing**. This study will be of interest to clinicians and biomedical researchers who work on microbiome and male fertility.

---

## [Referee Report · Reviewer #1 (Public review)]

Summary:

The authors analyzed the bacterial colonization of human sperm using 16S rRNA profiling. Patterns of microbiota colonization were subsequently correlated with clinical data, such as spermiogram analysis, presence of reactive oxygen species (ROS), and DNA fragmentation. The authors identified three main clusters dominated by Streptococcus, Prevotella, and Lactobacillus & Gardnerella, respectively, which aligns with previous observations. Specific associations were observed for certain bacterial genera, such as Flavobacterium and semen quality. Overall, it is a well-conducted study that further supports the importance of the seminal microbiota.

Strengths:

- The authors performed the analysis on 223 samples, which is the largest dataset in semen microbiota analysis so far

- Inclusion of negative controls to control contaminations.

- Inclusion of a positive control group consisting of men with proven fertility.

[Editors' note: the authors addressed the concerns raised in the previous round of review.]

---

## [Author Response]

The following is the authors’ response to the previous reviews.

Discussion: Could the authors discuss more the findings about Flavobacterium? Has it ever been associated with the urogenital tract?

Page 13-14, line 252-268:

*‘*The genus Flavobacterium was defined in 1923 to encompass gram-negative, non-spore-forming rods, of yellow pigment (44). The inclusiveness of this definition resulted in a collective of heterogenous species. By 1984 the genus had been restricted to those that were also non-motile and non-gliding (44). More recently, with an increase in genomic profiling, many species previously considered to be of genus Flavobacterium have been reclassified to genus Chryseobacterium, Cytophaga, and Weeksella (45). Increasing numbers of Flavobacterium species are being discovered such as gondwanense, Collinsii, branchiarum, branchiicola, salegens and scophthalmum (46) (47) (48). The allocation of Flavobacterium aquatile to this genus remains controversial due to its motility (49). Flavobacterium species are widely distributed in the environment including soil, fresh water and saltwater habitats (50) (51). There are many reports of pathogenic infections of Flavobacterium species in fish, however human infections are rare (48). A handful of case reports have described opportunistic infections to include pneumonia, urinary tract infection, peritonitis and meningitis (52) (53) (54) (55). Flavobacterium lindanitolerans and Flavobacterium ceti have been isolated as causative agents in some (56) (54). Case reports also describe Flavobacterium odoratum as a causative agent in urinary tract infection, most often in the immunocompromised or those with indwelling devices (57) (58) (59). However, this was one of many species previously of genus Flavobacterium reclassified, in this case to genus Myroides (60). Notably in our sample participants were asymptomatic of urinary tract infection’.

What is the relative abundance of Flavobacterium in the present study: this type of bacterium has been previously associated with contaminations (PMID: 25387460, 30497919).

Page 13, line 244-247:

‘The Flavobacterium genus taxon we identified as significantly associated with abnormal semen quality and sperm morphology was present in 36.28% of the samples, with a mean relative abundance of 1.15% in those samples. This information and the mention of previous findings of Flavibacterium in contamination studies have been added to the discussion’.

Figure 1: Increase the size of panel A.

Amended.

Figure 3: Can the authors indicate the relative abundance of each genus/species by the size of the node?

Co-occurrence network figure has been modified to display relative abundance of nodes.

Supplementary data: I don't see anywhere the decontam plots.

Decontam plots as suggested in the package vignette have been added in the GitHub repository. For practical purposes, the plot corresponding to the frequency testing only display a random subset (n=15) of the total taxa (n=82) flagged by this test as contaminants. The. .csv files with the outputs of each filter are available in the same directory

Line 12: Check the sentenceLine 15: Genera in italicsLine 33: Change "overall quality of the spermatozoa" to "overall semen quality"Lines 18-20: RephraseLine 87: 28F-BorreliaLine 134: "Seminal microbiota" or "Composition of the seminal microbiota"Line 159: "These included ... genera"Line 166: "Of note, Flavobacterium genus was..."Lines 187-188: Check sentence

Thank you, these have been amended